# Dynamics of Physiological Properties and Endophytic Fungal Communities in the Xylem of *Aquilaria sinensis* (Lour.) with Different Induction Times

**DOI:** 10.3390/jof10080562

**Published:** 2024-08-09

**Authors:** Qingqing Zhang, Rongrong Li, Yang Lin, Weiwei Zhao, Qiang Lin, Lei Ouyang, Shengjiang Pang, Huahao Zeng

**Affiliations:** 1Fujian Academy of Forestry, Fuzhou 350012, China; zqqteak@163.com (Q.Z.); lirr3635@163.com (R.L.); linyer99@163.com (Q.L.); leiouyang911@163.com (L.O.); 2School of Design, Fujian University of Technology, Fuzhou 350001, China; holyly@163.com; 3College of Forestry, Central South University of Forestry & Technology, Changsha 410004, China; zhao_vivi@163.com; 4Experimental Center of Tropical Forestry, Chinese Academy of Forestry, Pingxiang 536000, China; pangshengjiang@caf.ac.cn

**Keywords:** induction time, physio-biochemical properties, endophytic fungi diversity, network pattern, *Aquilaria sinensis*

## Abstract

Xylem-associated fungus can secrete many secondary metabolites to help *Aquilaria* trees resist various stresses and play a crucial role in facilitating agarwood formation. However, the dynamics of endophytic fungi in *Aquilaria sinensis* xylem after artificial induction have not been fully elaborated. Endophytic fungi communities and xylem physio-biochemical properties were examined before and after induction with an inorganic salt solution, including four different times (pre-induction (0M), the third (3M), sixth (6M) and ninth (9M) month after induction treatment). The relationships between fungal diversity and physio-biochemical indices were evaluated. The results showed that superoxide dismutase (SOD) and peroxidase (POD) activities, malondialdehyde (MDA) and soluble sugar content first increased and then decreased with induction time, while starch was heavily consumed after induction treatment. Endophytic fungal diversity was significantly lower after induction treatment than before, but the species richness was promoted. Fungal β-diversity was also clustered into four groups according to different times. Core species shifted from rare to dominant taxa with induction time, and growing species interactions in the network indicate a gradual complication of fungal community structure. Endophytic fungi diversity and potential functions were closely related to physicochemical indices that had less effect on the relative abundance of the dominant species. These findings help assess the regulatory mechanisms of microorganisms that expedite agarwood formation after artificial induction.

## 1. Introduction

Agarwood, an aromatic resin heartwood from the xylem tissues of trees in the Thymelaeaceae with the most widespread being *Aquilaria* trees [1], holds excellent economic value. It is extensively used in the manufacture of perfumes [2], ornamental displays [3], medicines [4,5] and toiletries [6]. Chinese medicine uses it as a natural sedative, carminative and anti-inflammatory agent [7]. Essential oils extracted from agarwood have been confirmed to have antimicrobial properties [8]. Agarwood production is considered a pathological process. Natural agarwood is extremely slow to form and infrequent; it occurs only when the stem of *Aquilaria sinensis* is exposed to external stress, triggering a series of defense reactions after injury, which is accompanied by changes in enzyme activities [9], phytohormone [10] and osmotic substance contents [11] as well as the diversity of endophytic microbial communities [12], ultimately resulting in the synthesis and accumulation of secondary metabolites. These metabolites of phenolic, chromones, terpenes substances and oils with aromatic odors eventually form agarwood through accumulation over the years [13]. As a result, natural agarwood has become extremely scarce, and wild resources have also been extensively illegally harvested; high market demand can no longer be met. To protect this species from extinction and achieve sustainable agarwood production, a large quantity of *Aquilaria* plantations have been established in recent decades across suitable growing areas with more than 80,000 hectares planted in China [14,15]. Consequently, the technology for artificially inducing agarwood has rapidly developed and has become a research hotspot.

Currently, exogenous interventions mainly consist of natural damage and artificial induction, such as lightning strikes and animal gnawing, as well as artificial physical damage, chemical stimulation and fungal infection. With *Aquilaria* plantations receiving increasing attention as a sustainable source of agarwood for various downstream industries, artificial induction techniques have been deeply explored in China and have spread globally to accelerate agarwood production for targeted downstream applications [16]. Numerous studies have shown that mechanical wounding approaches, including holing, fire drill and knife cutting, can promote agarwood production in *A. sinensis* trees [17,18,19] although with low yield and prolonged production timelines for agarwood of uncertain quality. Compared with conventional methods, the development of liquid induction approaches (applying inorganic salt solution, hormone liquid, fungal fermentation liquid, or their combination), known as the whole-tree agarwood-inducing technique, has greatly improved both yield and quality [20]. In practical applications, inducers such as inorganic salts or fungal liquids are more significant than mechanical injuries, and the coloration range and content of essential oil and alcohol soluble extract increase quickly [21,22]. Regardless of which induction stresses, trees accumulate malondialdehyde (MDA) and disrupt cellular metabolism in the initial stage as a result of increased membrane lipid peroxidation [23]. Trees undergo self-regulation through physiological and biochemical responses at the wound site in response to stress and protect themselves against pathogens. The activities of superoxide dismutase (SOD) and peroxidase (POD) also significantly enhance scavenging the excess reactive oxygen species (ROS), ensuring tree survival [24]. Concurrently, starch in the xylem is always consumed, and the soluble sugar content exhibits an increasing and then decreasing trend with resin formation [25,26,27]. The non-structural carbohydrates and enzyme activities mentioned above play important roles in agarwood formation, although their temporal dynamics under specific inductions remain unclear.

Endophytes are a group of microorganisms that can colonize healthy plant tissues, including the roots, stems, leaves, flowers and fruits, without causing any infection or diseases to the host [28,29]. They typically enter the host through lenticels, stomata, wounds or cut plant surfaces and establish a cooperative relationship with the host plants. Compared with pathogens, endophytes have been shown to decrease the expression of defense-related molecules when they invade a plant [30]. Endophytes primarily consist of endophytic fungi and bacteria, which are also well known for their functional attributes in increasing nutrient availability [31], encouraging host plant growth and development, alleviating abiotic stress on plants, alongside enhancing the resistance to phytopathogens to protect the host from biotic stress [32,33]. Of the various regulatory mechanisms, endophytic-mediated stress tolerance has been recognized by researchers as the most viable for sustained plant growth. Endophytes secrete a variety of metabolic substances within the host, most of which are anti-stress biochemicals [34,35]. In addition, endophytes can receive signals to activate stress-induced genes or activate genes associated with the tolerance mechanisms [36,37].

A growing number of studies have suggested that agarwood formation in *A. sinensis* is the result of a combination of external stresses and microorganisms, particularly endophytic fungi in the xylem [9,15,38]. Endophytic fungi from the surface-sterilized tissues of *A. sinensis* have been found to exhibit potent antimicrobial activity [39]. As essential drivers during agarwood formation, endophytic fungi produce a wide range of secondary metabolites and bioactive compounds with applications in perfumery and cosmetic industries [40,41]. Therefore, they are separated, identified and cultivated to produce high-quality agarwood by inoculation into *Aquilaria* trees [42,43]. Moreover, endophytic fungal communities constantly change in response to changes in host environment and stress intensity. The types and abundances of endophytic fungi have been distinguished before and after grafting the Qi-Nan clones [12]. The diversity and structure of endophytic fungi in the Qi-Nan clone have been shown to differ significantly with induction time, and a positive correlation between endophytic fungi and agarwood yield has been reported [44]. The process of agarwood formation has been synchronously detected as programmed cell death [45], which directly affects the ability of endophytic fungi to capture carbohydrates. Zhang et al. [46] concluded that wound induction promoted sesquiterpene biosynthesis and catheter obstruction without significant changes in the microbial community. Additionally, the results of induction vary depending on the circumstances, including the time and type of experiencing induction, and the species of the host plant. Thus, the link between endophytic fungi and plants is dynamic and complex under various induction conditions.

To better understand the role of endophytic fungi during the formation of agarwood, a ten-year-old *A. sinensis* plantation was used as the experimental material and was treated with inorganic salt inducers in this study. Xylem samples were collected at different induction times to determine their physiological properties and endophytic fungal species. The objectives of this study are to explore the dynamics of endophytic fungal communities in the xylem with induction time and to reveal the relationships between physiological properties and fungal structure.

## 2. Materials and Methods

### 2.1. Site Description and Plant Materials

The experimental site is located in Henghe Town, Huizhou City, southern China (114°6′ E, 23°20′ N) and has a subtropical monsoon climate with an average annual temperature of 21.8 °C. The extreme maximum and minimum temperatures are 37.9 °C and −2.4 °C, respectively. The climate is hot–rainy in summer and warm–dry in winter with an average annual precipitation of 1827 mm and 2050 h of sunshine. The site is situated at an altitude of approximately 60 m, and the soil is a granite-derived montane laterite. The plantation of *Aquilaria sinensis* was established in 2010 with an average diameter at breast height and tree height of 13.80 cm and 9.50 m before treatment, respectively.

### 2.2. Experimental Design

Healthy sample trees, free of visible disease and exhibiting generally uniform in growth were selected and treated in August 2020. There were four plots with 16 trees per plot. All trees were treated with a mixture of inorganic salt inducers consisting of 0.3% NaCl, 0.4% FeCl_2_ and 0.5% CaCl_2_. The specific operation method involved drilling four holes with a drill bit on the east, west, south, and north sides of the trunk positioned 50 cm above the ground. Subsequently, two bags of 250 mL mixed salt solution were injected into the trunk of each tree (Appendix A).

The xylem tissue of the stem was collected at four distinct time points: before treatment (0M) and at the third (3M), sixth (6M) and ninth months (9M) after artificial induction treatment. Samples were collected with a sterile knife and chisel from four trees per plot at 1.5 cm above and below the infusion holes. All samples were promptly frozen in liquid nitrogen after wrapping by sterile aluminum foil and stored at −80 °C for subsequent analysis of physiological characteristics and the endophytic fungal community.

### 2.3. Measurement of Physio-Biochemical Indices of Xylem Samples

Fresh samples were weighed and ground at a low temperature, and phosphate buffer was then added at a weight (g) to volume (mL) ratio of 1:15 in ice water. The mixture was centrifuged at 21,890× *g* for 10 min at 4 °C to obtain the supernatant, which served as a crude enzyme solution [9]. SOD and POD activities were determined using an enzyme activity test kit following the manufacturer’s instructions (Shanghai Enzyme-linked Biotechnology Co., Ltd., Shanghai, China). MDA content was determined using a thiobarbituric acid (TBA) colorimetric method [47]. Starch and soluble sugar contents were measured adopting to a spectrophotometer [48] and anthrone colorimetry [49], separately.

### 2.4. DNA Extraction and PCR Amplification of Xylem Endogenous Fungi

Xylem tissue samples were successively washed three times with 70% ethanol and 1X PBS solution and then ground into a powder with liquid nitrogen after drying. Subsequently, 0.5 g powder was placed in a 2 mL centrifuge tube for DNA extraction. Total community genomic DNA extraction was extracted using the E.Z.N.A.^TM^ Mag-Bind DNA kit (Omega, M5635-02, Omega Bio-Tek. Inc., Boston, MA, USA) according to the manufacturer’s instructions. To ensure the extraction of a sufficient amount of high-quality genomic DNA, the DNA integrity and concentration were monitored using a 1% agarose gel electrophoresis and a Qubit dsDNA HS test kit (Thermo Fisher Scientific Inc., Waltham, MA, USA), respectively.

The polymerase chain reaction (PCR) amplification of the fungal ITS1 region was performed using the primer sets ITS1F (5′-CTTGGTCATTTAGAGGAAGTAA-3′) and ITS2R (5′-GCTGCGTTCTTCATCGATGC-3′). PCR amplification was conducted in a PCR instrument (ETC811, Eastwin Life Sciences, Inc., Beijing, China) with a reaction system of 30 μL. The reaction was set up in sterile PCR tubes as follows: 2 μL of microbial DNA (10 ng/μL), 1 μL of amplicon PCR forward primer (10 μM), 1 μL of amplicon PCR reverse primer (10 μM), 15 μL of 2× Hieff^@^ Robust PCR Master Mix (Yeasen, 10105ES03, Yeasen Biotechnology Co., Ltd., Shanghai, China) and 11 μL of ddH_2_O. The thermal cycling program of the PCR reaction process was in a thermal instrument (Applied Biosystems 9700, Thermo Fisher Scientific Inc., Waltham, MA, USA) and included an initial denaturation at 95 °C for 3 min, first 5 cycles of denaturation at 95 °C for 30 s, annealing at 45 °C for 30 s, elongation at 72 °C for 30 s, then 20 cycles of denaturation at 95 °C for 30 s, annealing at 55 °C for 30 s, elongation at 72 °C for 30 s and a final extension at 72 °C for 5 min. PCR products were checked using electrophoresis on 2% (*w*/*v*) agarose gels in TBE (Tris, boric acid, EDTA) buffer stained with ethidium bromide (EB) and visualized under UV light.

### 2.5. Library Construction, Quantification and Sequencing

The free primers and primer dimer species present in the amplification product were purified using Hieff NGS^TM^ DNA Selection Beads (Yeasen, 10105ES03, Yeasen Biotechnology Co., Ltd., Shanghai, China). Before sequencing, the library concentration was determined using a Qubit^@^ 4.0 Green double-stranded DNA assay, and it was quality controlled using a bioanalyzer (Agilent 2100, Agilent Technologies, Inc., City of Santa Clara, CA, USA) to obtain uniform and high-quality sequencing data. Sequencing was performed by the Illumina MiSeq system (Illumina MiSeq, Illumina, Inc., San Diego, CA, USA) following established procedures. The raw data have been submitted to the Sequence Read Archive (SRA) data of the NCBI database with accession number PRJNA 1118912.

### 2.6. Bioinformatic Analyses of Sequencing Data

After sequencing, the two short Illumina readings were assembled by PEAR software (version 0.9.8 https://cme.h-its.org/exelixis/web/software/pear/ 20 March 2024) based on their overlap, and fastq files were processed to generate individual fasta and qual files. Effective tags were clustered into operational taxonomic units (OTUs) with a similarity of more than 97% using Usearch software (version 11.0.667 http://drive5.com/uparse/ 21 March 2024). The taxonomic classification of representative fungal OTU sequences was conducted by blasting against the UNITE fungal ITS Database (http://unite.ut.ee/index.php/ 21 March 2024) [50].

The α-diversity indices, including the Shannon diversity index, Chao1 richness index and Pielou’s evenness index, were quantified and calculated with Mothur software (version 3.8.31 http://mothur.org/ 25 March 2024). Beta diversity based on non-metric multidimensional scaling (NMDS) was analyzed using the ‘vegan’ package in R software (version 2.5-6, http://www.r-project.org/ 25 March 2024). The significance of clustering was evaluated by the utilization of similarity analysis (ANOSIM) [51]. The functional prediction of fungal OTUs was assigned using FUNGuild (v1.1) software [52].

### 2.7. Statistical Analysis

All data were first tested for normality and homogeneity before analysis of variance (ANOVA) employing SPSS25.0 statistical software (SPSS Inc., Chicago, IL, USA) and normalized using log_10_ (X + 1) transformations where necessary. One-way ANOVA and Tukey’s honestly significant difference (HSD) tests were conducted to analyze the discrepancy in physiological indicators, fungal diversity and species abundance at *p* < 0.05 among contrasting induction times. Based on the vegan package, principal coordinate analysis (PCoA) and ANOSIM were conducted to visualize and quantify dissimilarities in fungal potential functions. Subsequently, a Mantel test was carried out to identify the main drivers that were prominently correlated with the endophytic fungal community based on Pearson’s correlation (*p* < 0.05). All graphs were visualized in Origin Pro2024 software (Origin Lab Corp., Northampton, MA, USA).

Co-occurrence networks of fungi were constructed to explore the interactions among endophytic fungal species at different induction times. A Spearman correlation matrix was computed using the ‘psych’ package in RStudio software (2024.04.2+764), and values of r > |0.6| and *p* < 0.05 were set as thresholds for the construction of the modular network based on the order level. The visualization of fungal networks and calculation of topological properties were generated using the Gephi 0.9.1 software (http://gephi.org/ 10 May 2024) [53].

## 3. Results

### 3.1. Physio-Biochemical Properties of Xylem Samples

The enzyme activities, MDA levels and sugar content in the xylem of *A. sinensis* changed after artificial induction treatment and showed significant variation with the progression of induction time (Table 1). The activities of SOD and POD showed a tendency to increase and then decrease with induction time. Their activities were the strongest in the third month (3M) after induction and significantly enhanced by 52.56% and 38.69%, respectively, compared with the pre-induction (0M). Changes in MDA content coincided with enzyme activity with the maximum and minimum levels observed at 3M and 9M. In comparison to the pre-induction, starch content visibly declined after induction, whereas soluble sugar content accumulated to varying degrees.

### 3.2. Endophytic Fungal Community Diversity and Composition with Induction Time

A total of 2,825,832 raw data and 2,261,960 effective sequences of fungi were generated from the complete dataset of 16 xylem samples (Appendix A). The average number of sample sequences increased and then decreased with induction time, while the difference lacked significance (Figure 1A). All sequences were clustered into 819 OTUs with coverage above 99.50%; of these, 121 fungal OTUs were shared between four induction time groups. The number of unique OTUs was maximum at 9M, which was followed by 3M, 6M and 0M, and they tended to increase with induction time (Figure 1B).

#### 3.2.1. The Alpha Diversity of Endophytic Fungi

Significant differences in the xylem fungal communities were uncovered by comparing the diversity and richness indices at different induction times (Figure 1C). The number of OTUs increased significantly after artificial induction (in the 3M, 6M and 9M groups) compared with before the induction treatment (0M). The Shannon diversity index and Pielou’s index declined in the 3M and 6M groups and were lower than those in the 0M group, but conspicuousness was not observed between the 9M and 0M groups. The Chao1 richness index peaked in the 3M group and increased noticeably after induction compared with that before treatment.

#### 3.2.2. Composition of Endophytic Fungal Community

Nine different phyla, 32 classes, 87 orders, 179 families and 307 genera were obtained in all xylem samples of *Aquilaria sinensis* after definitive OTU annotation. The dominant orders (Figure 2A,B) and genera (Figure 2C,D) with an average relative abundance greater than 1% were analyzed.

At the order level, the number of dominant orders decreased after induction treatment and was 8, 7 and 13 in the 3M, 6M and 9M groups, respectively (Figure 2A). The top three orders in terms of relative abundance were also wildly distinct. The 0M group possessed 16 dominant orders; of these, Hypocreales (11.85%) was significantly dominant, which was followed by Conioscyphales (10.50%) and Calosphaeriales (10.20%). In contrast, the three most abundant genera in the 3M group were Polyporales (64.41%), Chaetothyriales (7.52%) and Pleosporales (7.47%). Eurotiales (58.56%), Chaetothyriales (16.20%) and Hypocreales (12.50%) were the three most dominant orders in the 6M group, and Chaetothyriales (24.55%), Eurotiales (13.25%) and Hypocreales (12.23%) were advantageous orders in the 9M group (Figure 2B).

The number of dominant genera and the top three genera with relative abundance varied obviously before and after artificial induction (Figure 2C,D). There were 18, 8, 4 and 14 genera with relative abundances higher than 1% in the 0M, 3M, 6M and 9M, respectively, showing that induction treatment significantly reduced the number of dominant genera. The top three genera in the 0M were Conioscypha (10.50%), Jattaea (10.20%) and Gibberella (8.48%). Candelabrochaete (64.41%) and Talaromyces (57.05%) exhibited the highest relative abundance at 3M and 6M, respectively, belonging to Basidiomycota and Ascomycota, respectively. Exophiala (13.55%), Talaromyces (12.28%) and Rhinocladiella (9.91%) were the three main genera in the 9M (Figure 2D).

### 3.3. Changes of Endophytic Fungal Community Structure

To compare the structure of endophytic fungal communities between different induction time groups, the NMDS based on Bray–Curtis distances (Figure 3A) and ANOSIM analysis based on weighted UniFrac distances (Figure 3B) were performed. NMDS analysis illustrated that the fungal β-diversity was clearly clustered into four groups according to the induction time. ANOSIM further confirmed that induction time significantly influenced the structure of the fungal community (R = 0.605, *p* = 0.001).

The relative abundances of Polyporales, Eurotiales, Chaetothyriales, Calosphaeriales, Chaetosphaeriales, Xylariales, Dothideales and Sordariales were significant different between induction times (Figure 4A,B, Appendix A). Chaetothyriales, Xylariales and Chaetosphaeriales significantly enhanced with induction time. An obvious decrease in the relative abundance of Calosphaeriales, Dothideales and Sordariales was displayed after induction compared with before induction. Polyporales and Eurotiales were predominantly enriched in the 3M and 6M groups, respectively.

Regarding the dominant genera of endophytic fungi, the differences in Candelabrochaete, Talaromyces, Jattaea, Rhinocladiella, Gibberella, Kurtzmanomyces, Phialemoniopsis, Aureobasidium, Nigrograna and Pyrigemmula were more obvious than those in other genera at different induction times (Figure 4C,D, Appendix A). Among these, Talaromyces, Rhinocladiella, Phialemoniopsis, Nigrograna and Pyrigemmula enriched clearly after induction treatment, whereas the average relative abundances of Gibberella and Aureobasidium reduced significantly. Candelabrochaete and Talaromyces were highly enriched int the 3M and 6M groups; they both belong to Basidiomycota and Ascomycota, individually.

### 3.4. Endophytic Fungal Co-Occurrence Network Analysis

Considering the significant effect we observed for before and after induction on the endophytic fungi in the xylem, co-occurrence networks were constructed among the annotated orders to elucidate co-occurrence patterns during agarwood formation (Figure 5). The network structures were appreciably different between pre- and after-induction with a more complex structure observed in the 9M group. The number of network nodes slightly increased from 46 to 64, and the edges greatly increased with induction time. Compared with 0M, network edge numbers were promoted by 12.22%, 67.22% and 143.33% in the 3M, 6M and 9M groups, respectively. These nodes mainly belonged to the phyla Ascomycota and Basidiomycota. Positive connections between orders were overwhelmingly dominant and exceeded 90% in the 0M, 3M and 9M groups (Figure 5A,B,D). Moreover, the average degree, network diameter and graph density also continuously heightened with induction time, reaching a maximum in the 9M group, and indicating the relationships between fungal species become more sophisticated in this network.

Before the induction of agarwood formation, the highly connected nodes of endophytic fungi were Sebacinales, Helotiales and Cystofilobasidiales, which are rare taxa (Appendix A). Within the 3M group, the primary central species were Pleosporales, Orbiliales and Botryosphaeriales with only the latter two being rare taxa. Nevertheless, the predominant taxa served as the main connectors in the network in the 6M and 9M groups (Appendix A). These findings indicate a gradual shift in keystone species within networks from rare to dominant taxa.

### 3.5. Predicted Functions of Fungal Communities

The predicted function result of the FUNGuild displayed 65 guilds belonging to seven trophic modes: saprotroph, pathotroph–saprotroph, pathotroph, pathotroph–saprotroph–symbiotroph, symbiotroph, pathotroph–symbiotroph and saprotroph–symbiotroph (Figure 6A). Among the identified fungal guilds, wood saprotrophs (average relative abundance 70.19%) were the most predominant guilds in the 3M group, undefined saprotrophs (57.48%), and animal pathogen–fungal parasite–undefined saprotrophs (23.48%) were the most predominant guild in the 6M and 9M groups, respectively (Figure 6B). However, the leading guilds in the pre-induction group were animal pathogen–endophyte–lichen parasite–plant pathogen–soil saprotroph–wood saprotroph (Figure 6B). Principal coordinate analysis (PCoA) based on the Bary–Curtis distance revealed that induction time significantly shaped different fungal functions (Appendix A). Similar to the discrepancy in fungal community composition, the function was much more sensitive to induction time (R = 0.438, *p* = 0.002, Appendix A).

### 3.6. The Correlation Analysis between Endophytic Fungi and Physio-Biochemical Indicators in the Xylem

Pearson correlation analysis between dominant orders abundances and physio-biochemical indices revealed that Sordariales, Saccharomycetales and Pezizales, which attached to Ascomycota, were markedly negatively correlated with SOD, POD and soluble sugar (Figure 7A). An obvious positive correlation was reflected between Pezizales and starch as well as Polyporales and soluble sugar. The correlation heat map showed that the relative abundances of Gibberella and Aspergillus were inversely correlated with SOD activity, whereas they were positively correlated with starch content (Figure 7B). POD activity was negatively correlated with the relative abundance of Gibberella, Aureobasidium, Echria and Chaetomium, and the analogous relationship was manifested among MDA and Rhinocladiella, Pyrigemmula; these species belonged to the Ascomycota. The soluble sugar content was positively correlated with the relative abundance of Candelabrochaete pertaining to Basidiomycota.

Mantel test results indicated an obvious positive correlation between fungal community α-diversity and potential function with SOD and POD activities as well as starch and soluble sugar content (Figure 8). Community composition was positively correlated with POD activity and soluble sugar content. Notably, the MDA content did not have a prominent impact on fungal diversity, community composition, or predicted function.

Comprehensive redundancy analysis (RDA) indicated that the starch and soluble sugar contents contributed more to endophytic fungal diversity (Appendix A). The potential function of endophytic fungi was mainly influenced by the SOD activity (Appendix A).

## 4. Discussion

### 4.1. Physiological Properties of Xylem with Induction Time

Abiotic or biotic stresses can produce excessive amounts of reactive oxygen species (ROS), which can lead to severe damage to plant cells [54,55]. Plants initiate a self-defense system by increasing the ratio and activity of antioxidant enzymes to scavenge excess superoxide anions to maintain ROS balance and mitigate cellular injury [56,57]. Furthermore, MDA is generated by the reaction of polyunsaturated fatty acids and oxygen species and can be widely used to evaluate oxidative damage in plant tissues [58]. In this study, an increased production of MDA was observed in the 3M and 6M groups after induction compared with pre-induction, indicating the presence of oxidative stress in *Aquilaria sinensis*. Correspondingly, SOD and POD activities in the xylem significantly boosted after chemical inducer treatment compared to the normal condition of *Aquilaria* trees. When the induction time prolongs, the activities of both tended to weaken, but they are still higher than the pre-induction level (Table 1). This result is basically consistent with a previous study showing that the antioxidant enzymes maintain higher activity during agarwood formation [9,25]. The higher MDA levels in the xylem suggest that induction treatment exacerbated lipid peroxidation in the initial period, thus prompting the plant to activate a complex stress response to mitigate damages by promoting SOD and POD activity. The decline in MDA content and enzyme activities may be attributed to xylem cell death and the formation of fragrance during the period from 6M to 9M [9,45]. Another explanation was also mentioned that the stress was attenuated over time without continuous interference, and *A. sinensis* are becoming adapted to the induction environment and repaired the injury [44,59].

In addition to these defensive enzymes, we also observed the dynamics of non-structural carbohydrates, which are essential for plant survival, development, tolerance, and defense under stressful status [60,61]. After the induction treatment of trees, starch levels in the xylem were constantly depleted, while the content of soluble sugar significantly accumulated more than before induction (Table 1). This indicates that external stress induces plants to convert starch into directly available sugars for participation in defense mechanisms. Previous studies have shown that non-structural carbohydrates provide a usable carbon source for agarwood production through the conversion of starch to soluble sugars and also accelerate the secretion of secondary metabolites [11]. The tendency of soluble sugar content to initially increase and then decrease with induction time was observed, which could be related to starch grains hydrolysis and the synthesis of sedum resins [22]. Initially, the starch grains in the cells were heavily hydrolyzed, leading to soluble sugar accumulation. However, with the agarwood resin production, cells with metabolic functions and the sieve tubes with transport functions in interxylary phloem were blocked and subsequently died due to an inadequate supply of energy. At this stage, parenchyma cells lose their ability to transport and break down the sugar compounds. Starch could not be metabolized to soluble sugars or transported and had to be temporarily retained in the resin-filled parenchyma cells [62,63]. Hence, the soluble sugar content in the xylem was decreased.

### 4.2. Diversity and Compositional Characteristics of Endophytic Fungi

The diversity and composition of the endophytic fungal community are predominantly affected by host types, growth stage and survival environment [12,64]. Our study showed a significant induction time-dependent variation in Shannon, Chao1 and Pielou’s indices of endophytic fungi in the xylem of *Aquilaria sinensis*. Specifically, the Shannon and Pielou indices showed a substantial decline in the early stage and subsequently elevated but did not return to the pre-induction level, whereas the Chao1 index maintained a high level (Figure 1C), suggesting that the fungal diversity decreased but the species richness promoted after induction treatment. Endophytes have developed two principal pathways to counteract abiotic stress: activating the host stress response and secreting resistant metabolites [32,65]. Tree damage leads to disruption of the microenvironment balance, causing a portion of the fungus to proliferate rapidly, which can increase the tolerance of plants exposed to salt stress. However, these reactions of plant-endophytes and high-salt environments possible directly inhibit the activity of some species [26,66]. This conclusion was also supported by the correlations between diversity and physio-biochemical indices. Similar results have been reported that fungal diversity is negatively related to the main components of incense such as sesquiterpenes and chromones [67]. Therefore, the diversity of fungal populations was declining, as verified by changes in Pielou’s indices. Moreover, the increase in species richness may have been caused due to the entry of fungi from the external environment through the infusion hole. Inconsistent results were reported by Li et al. [44], who observed a decline in community richness after induction treatment, which may be related to the differences in tree characteristics and induction approach.

In this study, 810 OTUs of endophytic fungi belonging to nine phyla were annotated from all xylem samples. Ascomycota and Basidiomycota were the two main dominant phyla, which is basically consistent with most research findings, indicating that Ascomycota is an extremely abundant group of saprophytic, promoting nutrient cycling in plants [68,69], and Basidiomycota is the main endophytic species that antagonizes the growth of pathogenic fungi that are harmful to plants [70]. These endophytic fungi exhibit a greater degree of coexistence owing to their stress tolerance [71]. The phenomenon in which consistent trends in the number of dominant orders, genera and Shannon index with induction time was observed, and the composition also has the distinction (Figure 2). Moreover, Xylariales, Chaetosphaeriales and Chaetothyriales were steadily enriched with induction time, and Talaromyces, Rhinocladiella, Phialemoniopsis, Nigrograna and Pyrigemmula also kept more higher abundance after induction (Figure 4). These dominant species may be associated with the inhibition of pathogens and the formation of agarwood resin. *Xylaria* sp. and *Rhinocladiella* spp. have been reported to be isolated and identified from agarwood and considered as one of the promising fungal species for the production of agarwood essential oil [72,73].

### 4.3. Change in Endophytic Fungal Community Structure and Function with Induction Time

In our study, endophytic fungal communities were clustered into four groups depending on the induction times (Figure 3). Additionally, the complexity of the co-occurrence networks gradually increased with induction time based on topological properties (Figure 5). This indicates that the community assembly of endophytic fungi in the xylem is prominently shaped by the induction time. Similarly, a study of the Qi-Nan clone showed a high degree of temporal heterogeneity in the community structure of endophytic fungi [44]. Evidence is growing that the properties of microbial ecological networks, which are used to illustrate interactions among co-existing or competing microorganisms, can influence the response of communities to microenvironmental changes [74]. Co-occurrence networks also can reveal more complexity than community composition in the resistance and resilience of microbial communities under stress conditions [75,76] as well as identify the highly connected taxa and keystone groups in the community [77]. With longer induction times, the nodes, edges, average degree and graph density increased steadily, demonstrating that the interactions among species continued to strengthen, and the community structure became more complex. Notably, a positive correlation among species was highlighted in the networks, which proved that endophytic fungi during agarwood formation were more mutualistic during agarwood formation. Consistent results have been reflected in the microbial networks of medicinal plants [68] We also found that the keystone species in the co-occurrence network were transformed from rare to dominant taxa with induction time, which may be related to the formation of incense resin.

According to the FUNGuild analysis, the endophytic fungal community differed significantly and was mainly composed of saprotrophs and pathotroph–saprotrophs, and the relative abundance of these groups was higher after induction than before induction (Figure 6 and Appendix A). A possible reason may be that the balance between the host and fungi is destroyed due to the induction treatment, causing endophytic fungi to shift toward saprophytic or pathogenic behavior [78]. Research has shown that saprotrophs and pathotrophs with saprophytic characteristics mainly obtain essential nutrients by degrading plant cells [68]. Correlation analysis also suggested that the relative abundance of functional groups was negatively correlated with SOD activity, starch content, and soluble sugar content (Figure 8). These findings suggest that endophyte assembly is closely related to the host environment, which is also the result of their combined actions.

## 5. Conclusions

Our study found that *A. sinensis* exhibited a strong stress response in the early stage of induction as evidenced by a promotion in resistance enzyme activity and starch consumption. The accompanying changes included alterations in the structure and function of the endophytic fungal communities in the xylem. With the induction time prolonged, species richness was continuously promoted, but the diversity first decreased and then increased. The interactions among fungal species intensified with dominant taxa occupying key positions instead of rare taxa. Correlation analysis and Mantel tests revealed a significant correlation among endophytic fungi, enzyme activity and carbohydrate content. Starch, soluble sugar and SOD were identified as the primary factors driving the community changes.

## Figures and Tables

**Figure 1 jof-10-00562-f001:**
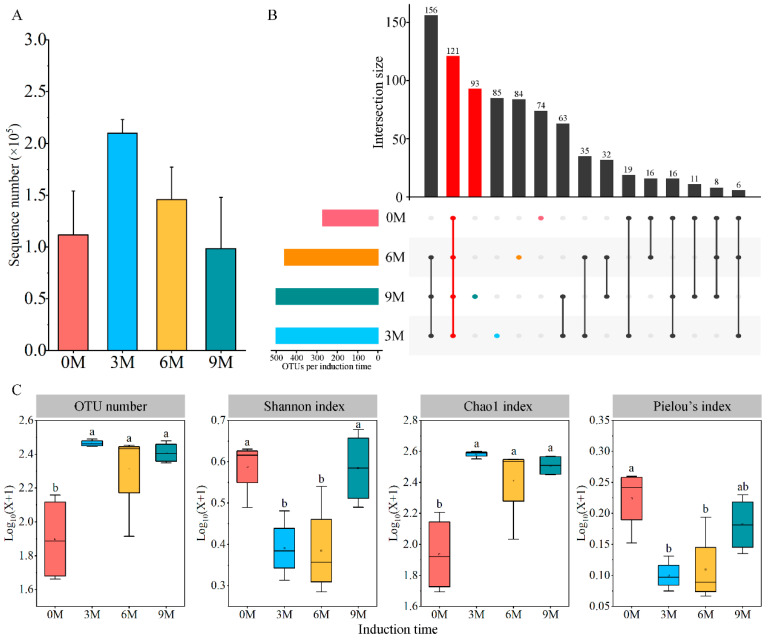
Number of sequences (**A**), unique OTUs (**B**) and diversity indices (**C**) of endophytic fungi at different induction times. 0M, 3M, 6M and 9M represent pre-induction, the third month, the sixth month and the ninth month after artificial induction, respectively. Different letters indicate significant differences between induction time groups (*p* < 0.05, Tukey’s test).

**Figure 2 jof-10-00562-f002:**
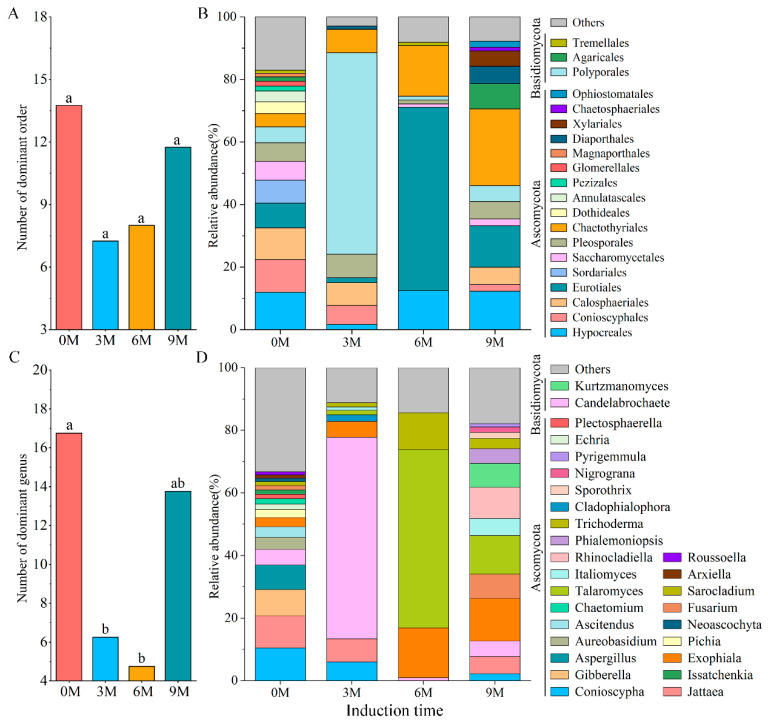
The number (**A**,**C**) and relative abundance (**B**,**D**) of dominant fungi at order (**A**,**B**) and genus (**C**,**D**) levels under different induction times.

**Figure 3 jof-10-00562-f003:**
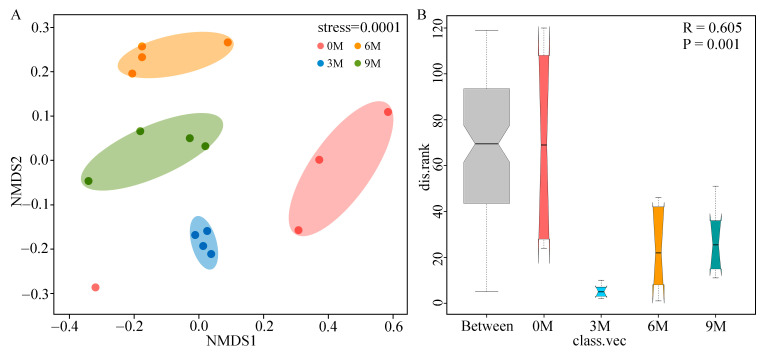
NMDS ((**A**), based on Bray–Curtis) and ANOSIM ((**B**), based on weighted UniFrac distances) analyses of endophytic fungi of *A. sinensis* at different induction times.

**Figure 4 jof-10-00562-f004:**
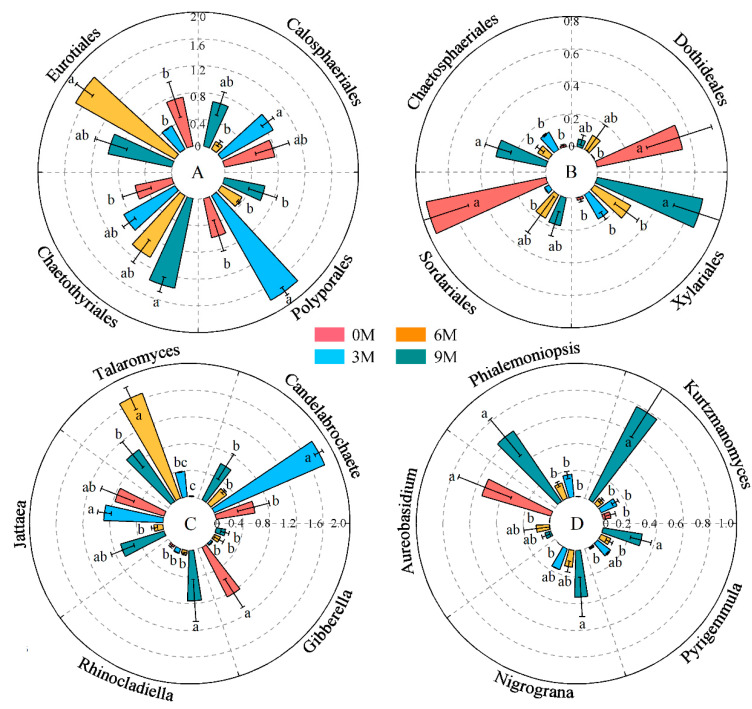
Changes in the relative abundance of dominant orders (**A**,**B**) and genera (**C**,**D**) in different induction times. The relative abundance of each indicator in the above figure was converted to log_10_(X + 1) standards. Different letters indicate significant differences between induction time groups (*p* < 0.05, Tukey’s test).

**Figure 5 jof-10-00562-f005:**
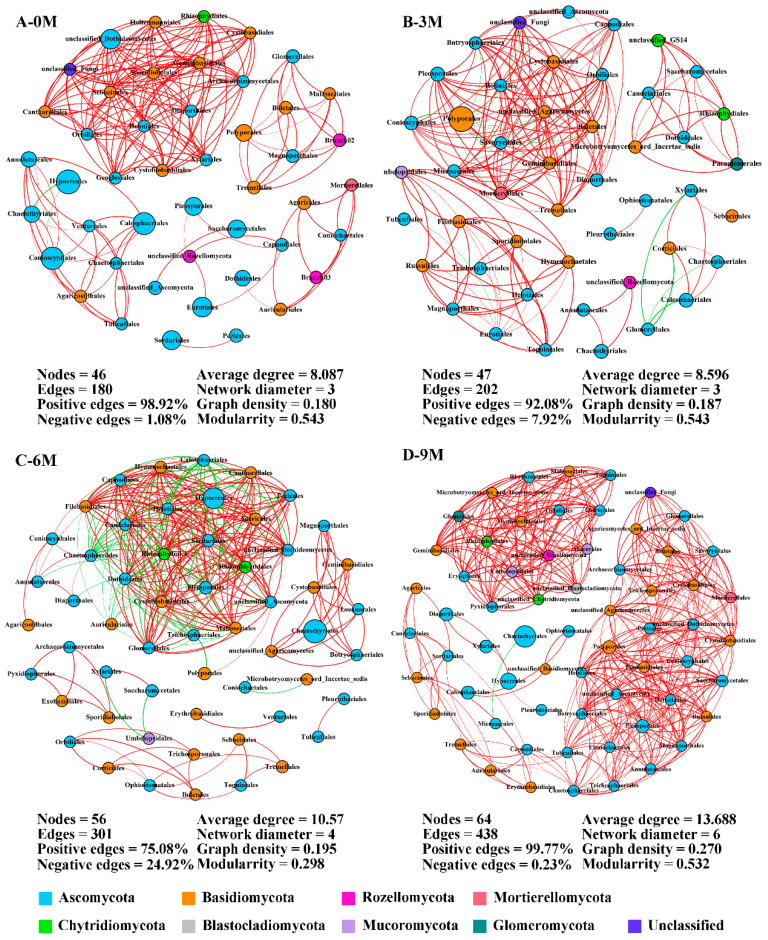
Co-occurrence network of endophytic fungi at the order level during different induction times. The size and color of each node depend on its abundance and phylum category. The red and green links indicate positive and negative correlations between nodes, respectively. 0M, 3M, 6M and 9M represent pre-induction, the third month, the sixth month and the ninth month after artificial induction, respectively.

**Figure 6 jof-10-00562-f006:**
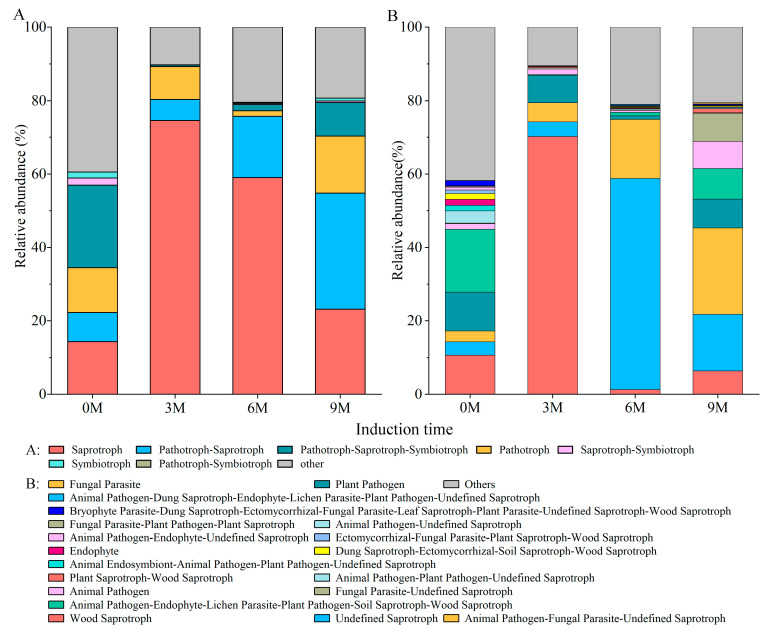
Function predicted of fungal community based on FUNGuild.

**Figure 7 jof-10-00562-f007:**
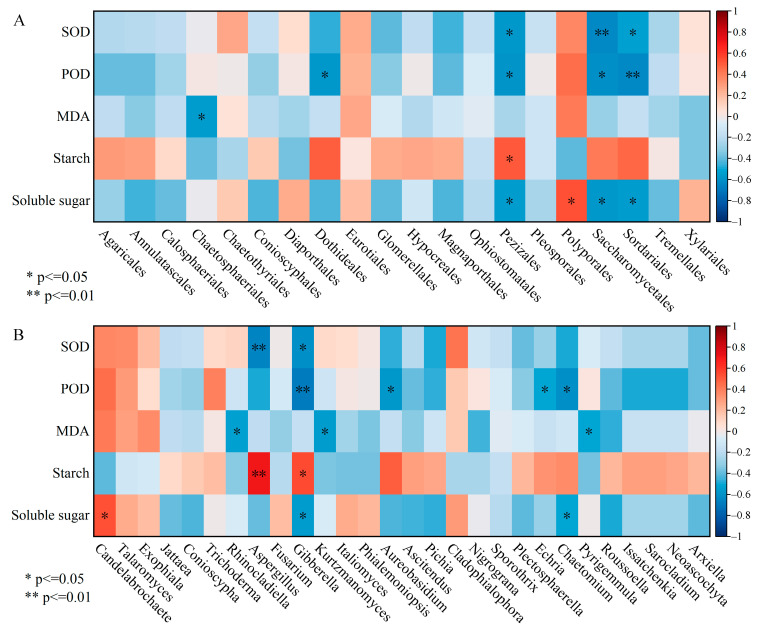
The correlation heatmap between dominant species (**A**: order level) (**B**: genus level) and physio-biochemical properties in the xylem. SOD, POD and MDA represent superoxide dismutase, peroxidase and malondialdehyde, respectively. * *p* < 0.05, ** *p* < 0.01.

**Figure 8 jof-10-00562-f008:**
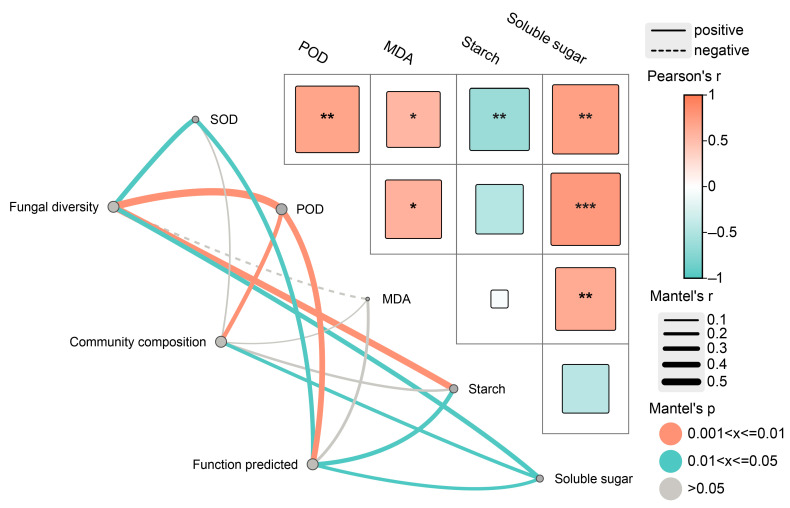
Mantel test among diversity, community composition, function predicted of endophytic fungi and physio-biochemical properties in the xylem. SOD, POD and MDA represent superoxide dismutase, peroxidase and malondialdehyde, respectively. * *p* < 0.05, ** *p* < 0.01, *** *p* < 0.001.

**Table 1 jof-10-00562-t001:** Physio-biochemical properties of *A. sinensis* xylem at different induction times.

Induction Time	Physio-Biochemistry Properties				
	SOD (U·g^−1^)	POD (U·mg^−1^)	MDA (nmol·g^−1^)	Starch Content (mg·g^−1^)	Soluble Sugar (mg·g^−1^)
0M	69.52 ± 4.71 b	8.53 ± 0.33 c	36.91 ± 1.42 bc	215.00 ± 4.96 a	23.53 ± 1.11 b
3M	106.06 ± 5.19 a	11.83 ± 0.30 a	44.57 ± 1.00 a	182.91 ± 5.31 b	33.78 ± 1.39 a
6M	104.74 ± 3.26 a	11.62 ± 0.36 a	43.93 ± 2.22 ab	198.02 ± 5.56 ab	32.17 ± 1.58 a
9M	86.20 ± 6.50 ab	10.10 ± 0.11 b	33.34 ± 2.01 c	184.93 ± 2.99 b	28.11 ± 1.91 ab

Notes: The values in the table were mean ± SD (n = 4). 0M, 3M, 6M and 9M represent pre-induction, the third month, the sixth month and the ninth month after artificial induction, respectively. SOD, POD and MDA represent superoxide dismutase, peroxidase and malondialdehyde, respectively. Different letters in the same column indicate significant differences (*p* < 0.05).

## Data Availability

Data will be made available on request. Raw sequence data were submitted to the NCBI Sequence Read Archive database (project accession number: PRJNA 1118912).

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
