# Peer review of "Dynamics of Physiological Properties and Endophytic Fungal Communities in the Xylem of Aquilaria sinensis (Lour.) with Different Induction Times"

_jof, 2024, doi:10.3390/jof10080562_

Round 1
Reviewer 1 Report
The research topic is relevant and has practical significance, as it is related to scientifically based technologies for whole-tree agarwood-inducing technique, which greatly improving both yield and quality of agarwood. The theoretical significance of the work is associated with the dynamics of endophytic fungal communities in the xylem with induction time and elucidating the relationships between physiological properties and structure of fungal community. The study corresponds to the theme of the Journal of Fungi. Undoubtedly, the article is of interest to ecologists and mycologists who study the biodiversity and functions of endophytic communities of woody plants, as well as to physiologists who trace the behavior of symbiotic systems under stress.
The manuscript is presented in a well-structured form. The cited references are mostly recent: the publications of the last 5 years account for 53%. The experimental design is suitable for testing the hypothesis. The "Methods" section is described in sufficient detail and clearly. Various software for bioinformatic analyses of sequencing data, taxonomic classification, biodiversity analysis and functional prediction of fungal OTUs is presented. Rich and well organized illustrative material in the “Results” section is clear, the data are interpreted appropriately and consistently throughout the manuscript. The conclusions are consistent with the evidence and arguments presented.
The article can be published in the journal after the clarifications, mentioned in the reviewer’s comments are made to the text.
There are 2 comments/questions about lines 147-152.
1) The authors call the applied effect chemical (use of inorganic salts). However, the method of introducing them into the tree trunk involves physical drilling. Would it not be worthwhile to provide additional control for the experimental design without the use of salts, only with mechanical drilling, in order to make sure that it was the chemical method of induction (not mechanical) that caused all the observed changes? The opinion of the authors should be indicated in the «Methods» section on this matter, based on their own research or other sources of information.
2) What determines the composition of salts? Have any preliminary studies been conducted and are there any references to them?
3) Lines 421-423: “The decline in MDA content and enzyme activities may be attributed to xylem cell death and the formation of fragrant during the period from 6M to 9M”. Can the authors provide photographs or links to data from other authors to confirm this assumption?
4) In a sentence on the lines 514-515: Correlation analysis also suggested that fungal functions were negatively correlated with SOD activity, starch content, and soluble sugar content (Figure 8), It’s not very clear what functions we are talking about.
Author Response
Response to Reviewer Comments
Firstly, we are very grateful to your review and valuable advices on this manuscript, which has greatly helped us to improve the quality of the manuscript. Secondly, the following are the answers and revisions we have made in response to review questions and comments. The modified sections are marked in blue background. The number in brackets indicate the line number of the modified content.
There are 2 comments/questions about lines 147-152.
1) The authors call the applied effect chemical (use of inorganic salts). However, the method of introducing them into the tree trunk involves physical drilling. Would it not be worthwhile to provide additional control for the experimental design without the use of salts, only with mechanical drilling, in order to make sure that it was the chemical method of induction (not mechanical) that caused all the observed changes? The opinion of the authors should be indicated in the «Methods» section on this matter, based on their own research or other sources of information.
Response: This question is worth considering in terms of what kind of control to adopt. We explain this as follows:
In this study, we sampled along an induced time gradient, where 0M was defined as before salt solution treatment. Wood samples without salt solution treatment were collected at 0M. Even when physical drilling was used as a control treatment, we also sampled immediately after drilling, these samples were no difference whether it was drilling treatment or non-drilling treatment.
In addition, based on our previous observations and research, at six and twelve months after the only drill treatment (using a 5 mm drill bit) for A. sinensis trees, we found that this part of the trunk (area beyond 1.0 cm from the drill hole) was still white wood and no incense substance had formed. We believed that drill treatment of this intensity had the same effect as no treatments. In this study, we collected samples 1.5 cm from the infusion hole. Therefore, we ignored the effect of drilling on tree trunks and used the no treatment instead of the drill treatment as a control.
Some studies concluded that physical drilling can also contribute to the formation of agarwood, and it is inconsistent with our observations, which were closely related to tree strain and treatment intensity (size and number of drill holes; knife cutting or fire drill, and so on).
2) What determines the composition of salts? Have any preliminary studies been conducted and are there any references to them?
Response:
Our team has conducted in-depth research on the induction of agarwood formation by exogenous substances, such as inorganic salt solution, hormones, fungi, exogenous gas, and combination. In one of our recent studies, different combinations of NaCl (0.3%, 0.6%, 0.9%), FeCl2 (0.2%, 0.4%, 0.6%) and CaCl2 (0.1%, 0.3%, 0.5%) were conducted to induce agarwood formation, and we found that the discoloration area and alcohol soluble extract content were better under the combined treatment with 0.3% NaCl, 0.4 FeCl2 and 0.5% CaCl2 (the paper related to this result has not been published yet). Therefore, we selected the combination treatment (0.3% NaCl, 0.4 FeCl2 and 0.5% CaCl2) to observe the dynamic changes in the endophytic fungal community.
Previous studies have shown that sodium, iron or calcium salts can induce the production of incense substances, according to the reference: 1) Thanh, L.V., Do, T.V., Son, N.H., et al. Impacts of biological, chemical and mechanical treatments on sesquiterpene content in stems of planted Aquilaria crassna trees. Agroforest Syst. 2015, 89:973-981. 2) Azren, P.D., Lee, S.Y., Emang, D., Mohamed, R. History and perspectives of induction technology for agarwood production from cultivated Aquilaria in Asia: a review. J. For. Res., 2019, 30:1-11.
3) Lines 421-423: “The decline in MDA content and enzyme activities may be attributed to xylem cell death and the formation of fragrant during the period from 6M to 9M”. Can the authors provide photographs or links to data from other authors to confirm this assumption?
Response:
Relevant references have been added in the revision. (Line 413)
We made this assumption based on the following two aspects, while the real reasons need to be explored further.
1) It is well known that the number of living cells in the xylem is drastically reduced from the white wood layer to the transition layer to the agarwood layer. According to the report by Liu et al [45], they believed that programmed cell death might be involved in the progress of wounding-induced agarwood formation, and more severe PCD corresponded to a higher rate of resin formation.
Reference 45: Liu, P.W.; Zhang, Y.X.; Yang, Y.; Lv, F.F.; Wei, J.H. Programmed cell death might involve in progress of wounding induced agarwood formation in stems of Aquilaria sinensis. Microsc. Res. Techniq. 2022, 85(8), 2904–2912.
2) According to the research reported by Liu et al [9], the activities of SOD and MDA in the xylem transition layer (TL) and agarwood layer (AL) first increased and then decreased with induction time. Moreover, the RDA results showed that the activities of SOD and MDA negatively correlated with the sesquiterpenoids, and negative correlations were also shown between MDA and 2-(2-phenylethyl) chromones.
Reference 9: Liu, T.F.; Liu, Y.X.; Fu, Y.L.; Qiao, M.J.; Wei, P.L.; Liu, Z.G.; Li, Y.J. Structural, defense enzyme activity and chemical composition changes in the xylem of Aquilaria sinensis during fungus induction. Ind. Crop Prod. 2024, 208, 117804.
4) In a sentence on the lines 514-515: Correlation analysis also suggested that fungal functions were negatively correlated with SOD activity, starch content, and soluble sugar content (Figure 8), It’s not very clear what functions we are talking about.
Response:
This sentence was not expressed clearly. “Fungal function“ has been replaced by “the relative abundance of functional groups“ (Line 503). According to different resource utilization methods, fungi are classified into three trophic mode groups in this study using FUNGuild: Pathotroph, Symbiotroph and Saprotroph, including 65 guilds which belonged to seven subgroups. We calculated the relative abundance of each functional group and analyzed their correlation with physio-biochemical properties.

Reviewer 2 Report
The submitted manuscript describes the change of fungal endophytic community in the xylem Aquilaria sinensis associated to the process of agarwood production induced by salt addition. Also, correlation between fungal endophytes community composition and biochemical properties of the agarwood was analyzed. The obtained results allowing to identify the fungal taxa associated with the inhibition of pathogens and the formation of agarwood resin. Such taxa include species within the orders Xylariales, Chaetosphaeriales and Chaetothyriales, Talaromyces, Rhinocladiella, Phialemoniopsis, Nigrograna and Pyrigemmula. Furthermore, the Xylaria sp. and Rhinocladiella spp. were identified as candidates for the production of agarwood essential oil.
The submitted manuscript is well written, the methodology is properly described, the results are clearly explained, and the discussion adequately addressed.
The subject of the submitted manuscript is of interest for mycologists, fungal ecologists, phytopathologists, and forest management scientists.
Thus, I consider that the document is suitable for its publication in the Journal of Fungi after minor corrections.
1. Line 163, please provide the centrifugation conditions in g forces, not in rpm. The rpm depends on the rotor and centrifuge used whereas g forces not.
2. In the Table 1, please leave only the title un the upper line of the table. Reallocate the rest of the text as note at the final line of the table.
3. Line 256, please replace the word “goods” by “good”.
4. Line 311, please replace “statistical differences” by “significantly different”.
5. The text in the figure 6B is too small, impossible to read. Please consider increasing the word size.
Author Response
Response to Reviewer Comments
Firstly, we are very grateful to your review and valuable advices on this manuscript, which has greatly helped us to improve the quality of the manuscript. Secondly, the following are the answers and revisions we have made in response to review questions and comments. The modified sections are marked in blue background. The number in brackets indicate the line number of the modified content.
1、Line 163, please provide the centrifugation conditions in g forces, not in rpm. The rpm depends on the rotor and centrifuge used whereas g forces not.
Response: Based on your comments, we have modified “rmp“ to be replaced by “g“. By reviewing our experiments records and the manufacturer’s instructions, we checked and corrected the centrifugation conditions. Line 154: The mixture was centrifuged at 21,890 ×g for 10 min at 4℃ to obtain the supernatant.
2、In the Table 1, please leave only the title un the upper line of the table. Reallocate the rest of the text as note at the final line of the table.
Response: we have placed the note contents at the final line of the table. (Line237-240)
3、Line 256, please replace the word “goods” by “good”.
Response: we have modified this sentence. (Line 246)
4、Line 311, please replace “statistical differences” by “significantly different”.
Response: we have replaced “statistical differences” with “significantly different”. (Line 301)
5、The text in the figure 6B is too small, impossible to read. Please consider increasing the word size.
Response: we have adjusted Figure 6. (Line 361)
Reviewer 3 Report
Fundamentally, the MS is fine in every respect.
One note: more enzymes involved in oxidative stress may have been included in the studies.
According to the nomenclature rules, they should be written in tilted (highlighted):
AD 450, 575, 652, 685, 693, 709
Extra space: 454
Ad 560: PhD dissertation?
Author Response
Response to Reviewer Comments
Firstly, we are very grateful to your review and valuable advices on this manuscript, which has greatly helped us to improve the quality of the manuscript. Secondly, the following are the answers and revisions we have made in response to review questions and comments. The modified sections are marked in blue background. The number in brackets indicate the line number of the modified content.
1、Major comments:
Fundamentally, the MS is fine in every respect.
One note: more enzymes involved in oxidative stress may have been included in the studies.
Response:
The process of plant antioxidant stress is extremely complex, and many enzymes participated in biochemical reactions, including not only SOD and POD, but also other enzymes. In this manuscript, we have selected two common enzymes for preliminary exploration, and further research will detect more biochemical indicators to reveal their physiological response mechanisms.
2、detail comments:
According to the nomenclature rules, they should be written in tilted (highlighted):
AD 450, 575, 652, 685, 693, 709
Extra space: 454
Ad 560: PhD dissertation?
Response:
According to the nomenclature rules, we have revised for writing in tilted in line 439, 565, 642, 675, 683, 699 and 443.
For Ad 560, this is a book, we have modified the reference format in line 549.
Reviewer 4 Report
The manuscript “Dynamics of Physiological Properties and Endophytic Fungal Communities in the Xylem of Aquilaria sinensis (Lour.) with Different Induction Times” is dedicated to evaluating the functions and regulatory mechanisms of microorganisms that accelerate agarwood formation in the xylem of Aquilaria sinensis (Lour.) after artificial induction. In my humble opinion, the work is well written and structured, with necessary and understandable Figures. The work is interesting from an ecological point of view (diversity of endophytic fungi, interactions between plants and microorganisms) and important from a practical point of view (agarwood is used in medicine and also in perfumery and cosmetology in China). Another importance of the work is related to the research of Aquilaria sinensis (Lour.) plantations, where an alternative source of agarwood is created, which could increase the availability of agarwood products on the market.
Minor points that should be clarified or improved.
1. In my humble opinion, the Abstract is too long, contains a lot of information, but the reader's attention is not drawn to the most important points. It would be good to shorten the text, emphasize the brightest and most important results obtained in the course of research and presented in detail in the manuscript.
2. Introduction, Lines 108-110: “A growing number of studies have suggested that agarwood formation in A. sinensis is the result of a combination of external stresses and microorganisms, particularly endophytic fungi in the xylem.” Please add references.
3. Materials and Methods, section 2.1. (Site Description and Plant Materials). It would be nice if the authors would add a photo of the plantation for the sake of clarity and to better immerse the reader in the subject matter.
4. Materials and Methods, Lines 155-156. How exactly were the xylem samples taken? With a sterile scalpel? Please give an explanation in the methods.
5. Results, Lines 292-294: “There were 18, 8, 4 and 14 genera with relative abundances higher than 1% in the 0M, 3M, 6M and 9M, respectively, showing a significant decreasing trend.” This does not look like a downward trend, because there is a clear increase in 9M. Please clarify.
6.Discussion, Line 413:”…..indicating the presence of oxidative stress in Aquilaria. sinensis.” Please remove the dot after Aquilaria; Line 415: ”….. compared to the normal condition of A. trees.” Please add the species epithet.
7. Discussion, Lines 464-465: “. Moreover, the increase in species richness may have been due to the entry of fungi from the external environment through the infusion hole.” How valid is the article's comparison of the true endophytic fungal communities in treatments before and after induction, assuming contamination?
The manuscript “Dynamics of Physiological Properties and Endophytic Fungal Communities in the Xylem of Aquilaria sinensis (Lour.) with Different Induction Times” is dedicated to evaluating the functions and regulatory mechanisms of microorganisms that accelerate agarwood formation in the xylem of Aquilaria sinensis (Lour.) after artificial induction. In my humble opinion, the work is well written and structured, with necessary and understandable Figures. The work is interesting from an ecological point of view (diversity of endophytic fungi, interactions between plants and microorganisms) and important from a practical point of view (agarwood is used in medicine and also in perfumery and cosmetology in China). Another importance of the work is related to the research of Aquilaria sinensis (Lour.) plantations, where an alternative source of agarwood is created, which could increase the availability of agarwood products on the market.
Minor points that should be clarified or improved.
1. In my humble opinion, the Abstract is too long, contains a lot of information, but the reader's attention is not drawn to the most important points. It would be good to shorten the text, emphasize the brightest and most important results obtained in the course of research and presented in detail in the manuscript.
2. Introduction, Lines 108-110: “A growing number of studies have suggested that agarwood formation in A. sinensis is the result of a combination of external stresses and microorganisms, particularly endophytic fungi in the xylem.” Please add references.
3. Materials and Methods, section 2.1. (Site Description and Plant Materials). It would be nice if the authors would add a photo of the plantation for the sake of clarity and to better immerse the reader in the subject matter.
4. Materials and Methods, Lines 155-156. How exactly were the xylem samples taken? With a sterile scalpel? Please give an explanation in the methods.
5. Results, Lines 292-294: “There were 18, 8, 4 and 14 genera with relative abundances higher than 1% in the 0M, 3M, 6M and 9M, respectively, showing a significant decreasing trend.” This does not look like a downward trend, because there is a clear increase in 9M. Please clarify.
6.Discussion, Line 413:”…..indicating the presence of oxidative stress in Aquilaria. sinensis.” Please remove the dot after Aquilaria; Line 415: ”….. compared to the normal condition of A. trees.” Please add the species epithet.
7. Discussion, Lines 464-465: “. Moreover, the increase in species richness may have been due to the entry of fungi from the external environment through the infusion hole.” How valid is the article's comparison of the true endophytic fungal communities in treatments before and after induction, assuming contamination?
Author Response
Response to Reviewer Comments
Firstly, we are very grateful to your review and valuable advices on this manuscript, which has greatly helped us to improve the quality of the manuscript. Secondly, the following are the answers and revisions we have made in response to review questions and comments. The modified sections are marked in blue background. The number in brackets indicate the line number of the modified content.
Minor points that should be clarified or improved.
1、In my humble opinion, the Abstract is too long, contains a lot of information, but the reader's attention is not drawn to the most important points. It would be good to shorten the text, emphasize the brightest and most important results obtained in the course of research and presented in detail in the manuscript.
Response: we have made modifications to the Abstract. (Line 16-32)
Xylem-associated fungus can secrete many secondary metabolites to help Aquilaria trees resist various stresses and play a crucial role in facilitating agarwood formation. However, the dynamics of endophytic fungi in Aquilaria sinensis xylem after artificial induction have not been fully elaborated. Endophytic fungi communities and xylem physio-biochemical properties were examined before and after induction with an inorganic salt solution, including four different times (pre-induction (0M), the third (3M), sixth (6M) and ninth (9M) month after induction treatment). The relationships between fungal diversity and physio-biochemical indices were evaluated. The results showed that superoxide dismutase (SOD) and peroxidase (POD) activities, malondialdehyde (MDA) and soluble sugar content first increased and then decreased with induction time, while starch was heavily consumed after induction treatment. Endophytic fungal diversity was significantly lower after induction treatment than before, but the species richness was promoted. Fungal β-diversity was also clustered into four groups according to different times. Core species shifted from rare to dominant taxa with induction time, and growing species interactions in the network indicate a gradual complication of fungal community structure. Endophytic fungi diversity and potential functions were closely related to physicochemical indices that had less effect on the relative abundance of the dominant species. These findings help assess the regulatory mechanisms of microorganisms that expedite agarwood formation after artificial induction.
2、Introduction, Lines 108-110: “A growing number of studies have suggested that agarwood formation in A. sinensis is the result of a combination of external stresses and microorganisms, particularly endophytic fungi in the xylem.” Please add references.
Response: The references have been added in Line 100.
3、Materials and Methods, section 2.1. (Site Description and Plant Materials). It would be nice if the authors would add a photo of the plantation for the sake of clarity and to better immerse the reader in the subject matter.
Response: This suggestion is worthy of being considered. We have added a photo of the plantation in supplementary materials.
4、Materials and Methods, Lines 155-156. How exactly were the xylem samples taken? With a sterile scalpel? Please give an explanation in the methods
Response: Sampling methods have been added to the manuscript. Line 146: samples were collected with a sterile knife and chisel from four trees per plot at 1.5 cm above and below the infusion holes
5、Results, Lines 292-294: “There were 18, 8, 4 and 14 genera with relative abundances higher than 1% in the 0M, 3M, 6M and 9M, respectively, showing a significant decreasing trend.” This does not look like a downward trend, because there is a clear increase in 9M. Please clarify.
Response: We recognized your question very much and also have realized that there is considerable imprecision in the terminology used, which has been carefully revised. Line 284: There were 18, 8, 4 and 14 genera with relative abundances higher than 1% in the 0M, 3M, 6M and 9M, respectively, showing that induction treatment significantly reduced the number of dominant genera.
6、Discussion, Line 413:”…..indicating the presence of oxidative stress in Aquilaria. sinensis.” Please remove the dot after Aquilaria; Line 415: ”….. compared to the normal condition of A. trees.” Please add the species epithet.
Response: The dot has been removed. ‘Aquilaria’ has replaced ‘A.’ in line 405.
7、Discussion, Lines 464-465: “. Moreover, the increase in species richness may have been due to the entry of fungi from the external environment through the infusion hole.” How valid is the article's comparison of the true endophytic fungal communities in treatments before and after induction, assuming contamination?
Response: Firstly, completely sterile conditions are difficult to achieve in field experiments. Even though it is sterile in the early stages of processing, this state can be broken over time due to various factors. Here, we are only guessing that external fungus might entry the trunk. Secondly, if this assuming contamination was valid, we believe that the validity of the comparison results can be maintained. The fungal community in the xylem is mainly composed of endophytic fungi, and external contamination is only a very small part and their transfer in the xylem is also limited.
